

# Effects of ferulic acid esterase-producing *Lactobacillus fermentum* and cellulase additives on the fermentation quality and microbial community of alfalfa silage

Rina Su[1,3,*], Kuikui Ni[2,*], Tianwei Wang[1,3], Xiaopan Yang[1,3], Jie Zhang[3], Yayong Liu[1,3], Weixiong Shi[1,3], Liu Yan[4], Chen Jie[4] and Jin Zhong[1,3]

[1] School of Life Science, University of Chinese Academy of Sciences, Beijing, China
[2] College of Grassland Science and Technology, China Agricultural University, Beijing, China
[3] State Key Laboratory of Microbial Resources, Institute of Microbiology, Chinese Academy of Sciences, Beijing, China
[4] Hebei Zhongyu Zhongke Biotechnology Development Company, Hebei, China
[*] These authors contributed equally to this work.

## ABSTRACT

**Background.** Alfalfa (*Medicago sativa*) is an important forage material widely used for animal feed production. Ensiling is an effective method for preserving alfalfa, but it has shown some limitations in the production of high-quality alfalfa silage due to its low water soluble carbohydrates (WSC) content and high buffering capacity. Lactic acid bacteria (LAB) and cellulase are often used as silage additives to promote the ensiling process and enhance fermentation quality.

**Methods.** Experiments were conducted to investigate the effects of ferulic acid esterase (FAE)-producing *Lactobacillus fermentum* 17SD-2 (LF) and cellulase (CE) on the fermentation quality and microbial community of alfalfa silage. After 60 days of ensiling, analysis of fermentation quality and bacterial diversity in alfalfa silages were conducted using high-performance liquid chromatography and high-throughput sequencing methods.

**Results.** Alfalfa was ensiled with additives (LF, CE, and LF+CE) or without additives for 60 days. All additives increased lactic acid and decreased pH values and ammonia-N contents compared to control. Among all treatments, the combined addition of LF and CE showed lowest pH (4.66) and ammonia-N (NH$_3$-N, 0.57% DM) content, highest contents of lactic acid (LA, 10.51% DM), dry matter (DM, 22.54%) and crude protein (CP, 24.60% DM). Combined addition of LF and CE performed better in reducing neutral detergent fiber (NDF, 29.76% DM) and acid detergent fiber (ADF, 22.86% DM) contents than the addition of LF (33.71, 27.39% DM) or CE (32.07, 25.45% DM) alone. Moreover, the microbial analysis indicated that LF+CE treatments increased the abundance of desirable *Lactobacillus* and inhibited the growth of detrimental *Enterobacter* and *Clostridia* in alfalfa silage.

**Discussion.** Combined addition of FAE-producing LF and CE is more effective than treatments of LF or CE alone in improving fermentation quality and nutrition values of alfalfa silage. This is likely due to a synergistic effect of CE and FAE produced by LF on plant cell wall degradation, indicating that these additives promote each other to

Corresponding author
Jin Zhong, zhongj@im.ac.cn

improve fiber degradation and silage fermentation. In conclusion, combined addition of FAE-producing LF and CE could be a feasible way to improve alfalfa silage quality.

## INTRODUCTION

Alfalfa is widely used for animal feed in the world due to its high protein content. However, the low concentration of WSC and high buffering capacity in alfalfa makes it difficult to ensile for high silage quality (*Silva et al., 2016*). In order to enhance fermentation quality, lactic acid bacteria (LAB) and cellulase are often used as silage additives. LAB can accelerate lactic acid fermentation and reduce nutrition loss during the ensiling process (*Gulfam et al., 2017*). Cellulase can degrade the plant cell wall and improve the ruminal digestibility of silage (*Khota et al., 2016*).

Ferulic acid ester usually links ferulic acid with polysaccharides in plant cell walls, which results in extended networks and forms a protective layer. Ferulic acid esterase (FAE) can break the ester linkage between the ferulate and polysaccharide chain, and then enhance the accessibility of cellulase and hydrolyze the cell wall (*Koseki et al., 2009*). Well-characterized FAEs are mainly from fungi (*Faulds et al., 2005*), but now some researchers are increasingly studying FAE-producing bacteria due to their practical application value (*Fritsch et al., 2017*).

For silage fermentation, Jin et al. reported that mixed small-grain silage treated with FAE-producing *Lactobacillus buchneri* had higher *in situ* NDF digestibility than control. Furthermore, pretreatment with *L. plantarum* A1 (FAE activity) alone or in combination with exogenous cellulase had a synergistic effect on corn stalk silage fermentation and degradation of lignocellulase (*Li et al., 2019*), such as producing more available substrates for LAB fermentation. However, the effects of FAE-producing LAB inoculant on silage quality can be highly variable, possibly due to the different forage varieties and performance differences in LAB strains (*Lynch, Baah & Beauchemin, 2015*; *Jin et al., 2017*; *Li et al., 2019*).

The potential of FAE activity LAB to improve silage quality or fiber digestibility of various silages includes barley, ryegrass and corn have been reported (*Addah et al., 2012*; *Nsereko et al., 2008*; *Kang et al., 2009*). But there is little information about the application of FAE activity LAB or its combination with exogenous cellulase on alfalfa silage. We hypothesized that the combination of FAE-producing LF and CE will improve the fermentation quality and nutrition values more effectively in alfalfa silage. In order to verify the hypothesis, experiments were carried out as follows: characterize FAE activity *L. fermentum* 17SD-2; investigate the effects of *L. fermentum* 17SD-2 (LF), cellulase (CE), or their combination on the fermentation quality and bacterial community of alfalfa silage.

## MATERIALS AND METHODS

### Identification of FAE activity in *Lactobacillus fermentum* 17SD-2

*Lactobacillus fermentum* 17SD-2 used in this study was isolated from maize silage without any additives, then maintained in De Man, Rogosa, and Sharpe (MRS) broth containing 20% (v/v) glycerol at −80 °C. Before any experiments, the strain was propagated twice and cultured in MRS broth for 24 h at 37 °C, harvested by centrifugation (10,000 rpm, 2 min), and washed twice with phosphate buffered saline (PBS, pH 7.0). The sediment was resuspended in sterile water and 10 µL of the suspension was transformed to MRS plates with glucose omitted but containing 1g/L ethyl ferulate and incubated for 72 h at 37 °C, and *Lactobacillus fermentum* 5007 (CGMCC 1.3223) was used as negative control because it was previously confirmed to have no FAE activity. After incubation, we observed whether a clear zone formed around the suspension. If a clear zone appeared on the MRS plates, this indicated that the strain *L. fermentum* 17SD-2 had FAE activity.

### Morphological, physiological, and biochemical characteristics of *Lactobacillus fermentum* 17SD-2

Gram staining and catalase activity of *L. fermentum* 17SD-2 were determined after 24 h of incubation on MRS agar or broth. To measure the growth curve of *Lactobacillus fermentum* 17SD-2, the bacterial cells at a density of $1 \times 10^7$ CFUs/ml were seeded in MRS broth. The absorbance of *Lactobacillus fermentum* 17SD-2 culture at 600 nm were regularly monitored with a microplate reader. To evaluate salt tolerance, 2% culture was inoculated into MRS broth containing 3.0% and 6.5% NaCl, and then cultured at 37 °C for 24 h. To evaluate acid tolerance, 2% culture was inoculated into MRS broth with pH 3.5, 4.0, 4.5, 8.0, 8.5, and 9.0, and then cultured at 37 °C for 24 h. To evaluate temperature tolerance, 2% culture was inoculated into MRS broth, and then cultured at different temperatures (including 15, 20, 25, 30, 35, 40, and 45 °C) for 24 h. Then the absorbance of *Lactobacillus fermentum* 17SD-2 cultures at 600 nm was measured with a microplate reader. The value of $OD_{600}$ greater than 0.3 was considered as positive, expressed with "+". API 50CH contains a high level of carbohydrates and was used to determine carbon source utilization according to the manufacturer instructions.

### Alfalfa harvest and silage preparation

Alfalfa at blooming stage was manually harvested from an experimental field with an area of 5 m × 10 m, located at the Wuqing District (117°10′E, 39°10′N), Tianjin, China on June 7th, 2018. At first, the harvested alfalfa was divided into two groups: fresh (sample1, S1) and wilted (sample 2, S2, wilted in the field for 5 h). Alfalfa with various dry matter (DM) contents (S1: 23.82%; S2: 29.63%) was directly chopped into pieces in length of 2–3 cm using a crop chopper.

A commercial CE (10,000 U/g; Macklin Biochemical Co., Ltd, Shanghai, China) and *L. fermentum* 17SD-2 were used as additives for silage making. *L. fermentum* 17SD-2 could grow well at low pH conditions and possess high FAE-activity (clear zone with a diameter of 11.3 mm). The inoculant was added at a level of $10^6$ colony forming unit (CFU) per gram of fresh matter (FM). CE was applied at a ratio of 1 g kg$^{-1}$ of FM (*Chen et al., 2017*).

The experimental treatments were designed as follows. Each group (sample 1 and sample 2) included 12 piles (500 g per pile), and they were randomly ensiled with four additives treatments: (i) untreated control (CK); (ii) application of commercial cellulase (CE); (iii) application of *L. fermentum* 17SD-2 (LF); and (iv) combination of *L. fermentum* 17SD-2 and commercial CE (LF+CE). Then, chopped material was mixed homogenously with the additives and packed manually into 35 cm × 50 cm polyethylene bags which were tightly vacuumed, and triplicates for per treatment were prepared. A total of 24 bags (2 groups × 4 treatments × 3 replicates) were kept at ambient temperature (21–30 °C). After 60 days of ensiling, three bags per treatment were opened to evaluate their fermentation end products, chemical composition, and microbial communities.

## Microorganism and fermentation quality analysis

To evaluate microbial counts, 10 g forage (raw materials and silage) was extracted with 90 mL 0.85% sterile physiological saline solution, and serially diluted from $10^{-1}$ to $10^{-6}$. In total, 100 μL from an appropriate dilution was spread on agar plates. Lactic acid bacteria were enumerated on MRS agar after incubation at 30 °C for 72 h. Molds and yeasts were counted on potato dextrose agar (Nissui) after incubation at 28 °C for 24 h, and yeasts were distinguished from molds or other bacteria by colony appearance and cell morphology observation (*Avila et al., 2009*). Plates containing a minimum of 30 and a maximum of 300 colony-forming units were enumerated (*Reich & Kung, 2010*). To determine the fermentation parameters, the sample (25 g) was mixed with 225 mL sterile water and incubated at 4 °C overnight and then filtered through four layers of cheesecloth (*Touno et al., 2014*). The filtrate was used to measure pH and the concentrations of ammonia-N and organic acid. The ammonia-N content was determined according to phenol-hypochlorite and ninhydrin colorimetric procedures described in a previous study (*Broderick & Kang, 1980*). Organic acids were analyzed using high-performance liquid chromatography (HPLC), equipped with a UV detector and set as follows: ICSep COREGEL-87H column, eluent five mmol/L $H_2SO_4$ with a running rate of 0.6 mL/min, temperature of column oven 50 °C, injection volume of 10 μL.

## Analysis of chemical composition

To determine the DM content, the samples were dried in a forced-air oven at 65 °C for 48 h and then dried at 103 °C to constant weight. The silage DM losses during drying were calculated according to the formula of *Porter et al. (1995)*. Dried sample was grinded to 1.0 mm particle diameter. The WSC was determined using the anthrone method (*Murphy, 1958*) and the CP was analyzed according to the standard procedure detailed by the Association of Official Analytical Chemists (*AOAC, 1990*). The content of NDF and ADF were measured according to the method described in a previous study (*Van Soest, Robertson & Lewis, 1991*).

## Bacterial diversity analysis

Samples (20 g) were mixed with 180 mL of sterile 0.85% NaCl solution with vigorous shaking at 120 r/m for 2 h. The mixture was filtered through four layers cheesecloth and the filtrate was centrifuged at 10,000 r/m for 10 min at 4 °C. The deposit was resuspended

in one mL of sterile 0.85% NaCl solution and the microbial pellets were obtained by centrifugation at 12, 000 r/m for 10 min at 4 °C. To extract total DNA in raw material and silage samples (30 samples in total), the DNA Easy Power Soil Kit (Qiagen, Hilden, Germany) was used according to the manufacture's protocols. The PCR reactions were conducted in a 20 μL mixture (10 ng of template DNA, 2 μL of 2.5 mM dNTPs, 0.8 μL of each primer (5 μM), 0.4 μL of FastPfu Polymerase, and 4 μL of 5×FastPfu Buffer, 0.2 μL of BSA, 11.8 μl of ddH$_2$O). According to *Ni et al. (2017)*, the 16S rDNA V3-V4 regions were amplified using primers 338F (ACTCCTACGGGAGGCAGCAG) and 806R (GGACTACHVGGGTWTCTAAT). Triplicate PCR reaction for each sample was conducted to minimize PCR deviation, and a mixture of the three PCR products was sequenced.

## High-throughput sequencing and bacterial diversity analysis

Amplicons were extracted from 2% agarose gels, purified with a Qiagen Gel Extraction Kit (Qiagen, Hilden, Germany) and sequenced at the Shanghai Majorbio Bio-pharm Technology Co., Ltd using paired-end sequencing (2×300 bp) with an Illumina MiSeq platform according to the standard protocols. Raw tags were quality-filtered by Trimmomatic (Version 3.29) and merged by FLASH (Version 1.2.7) with the following criteria: (i) the reads were truncated at any site receiving an average quality score <20 over a 50 bp sliding window; (ii) sequences with overlaps longer than 10 bp were merged according to their overlap with mismatches no more than 2 bp; and (iii) sequences of each sample were separated according to barcodes (exactly matching) and primers (allowing 2 nucleotides mismatching) and reads containing ambiguous bases were removed. Operational taxonomic units (OTUs) were clustered with 97% similarity using UPARSE (version 7.1, http://drive5.com/uparse/). OTUs were used to calculate rarefaction and alpha diversity (Mothur (v.1.30.1).

## Statistical analysis

Data shown are means ± standard deviation (SD). All microbial counts data were converted to log10 and the results were described on a fresh weight basis. Two-way analysis of variance was used to evaluate the effects of water contents, additives and their interaction on microbial population, pH values, fermentation characteristic, chemical composition and alpha diversity of bacterial community in alfalfa silage. All the statistical analyses were performed using the general linear model procedure of SAS (version 9.0, 2002; SAS Institute, Cary, NC, USA). The means were compared for significance by Duncan's multiple range method and the significance was declared at $P < 0.05$.

## RESULTS

### Chemical and microbial composition of raw materials and the characteristics of *Lactobacillus fermentum* 17SD-2

The chemical composition and microbial populations of the alfalfa crops before ensiling are shown in Table 1. The DM contents of sample 1 and sample 2 were 23.82% and 29.63% of FM, respectively; their CP contents were 22.63 and 25.34% DM, respectively; and NDF and

**Table 1  Chemical composition, cultivable microbial population and alpha diversity of bacterial community of raw materials before ensiling.**

| Item | S1 | S2 | *P* value |
|---|---|---|---|
| Chemical composition | | | |
| DM% | 23.81 ± 1.12 | 29.63 ± 0.53 | 0.0012 |
| CP (%DM) | 22.62 ± 0.35 | 25.34 ± 0.21 | 0.0003 |
| NDF (%DM) | 38.39 ± 0.02 | 33.35 ± 0.28 | 0.00001 |
| ADF (%DM) | 29.73 ± 0.05 | 25.54 ± 0.4 | 0.0001 |
| WSC (%DM) | 1.54 ± 0.12 | 1.8 ± 0.05 | 0.0219 |
| Cultivable microbial population | | | |
| Yeast (log10 cfu/g FM) | 4.38 ± 0.03 | 4.08 ± 0.01 | 0.00005 |
| Mold (log10 cfu/g FM) | 4.2 ± 0.01 | 3.9 ± 0.12 | 0.0099 |
| Alpha diversity of bacterial community | | | |
| Observed species | 249.67 ± 17.21 | 175 ± 19.29 | 0.0075 |
| Shannon | 1.94 ± 0.46 | 1 ± 0.02 | 0.0244 |
| Simpson | 0.33 ± 0.05 | 0.61 ± 0.02 | 0.0006 |
| ACE | 327.64 ± 61.36 | 262.24 ± 15.8 | 0.1483 |
| Chao 1 | 309.88 ± 28.96 | 241.81 ± 7.24 | 0.0168 |
| Coverage | >0.99 | >0.99 | NA |

**Notes.**

FM, fresh material; DM, dry matter; CP, crude protein; NDF, neutral detergent fiber; ADF, acid detergent fiber; WSC, water soluble carbohydrates; LAB, lactic acid bacteria; CFU, colony forming units; NA, not applicable.

ADF concentrations were 33.3–38.3% and 25.5–29.7% DM, respectively. In this study, the WSC contents were lower than 1.8% of DM in both samples. Additionally, the numbers of epiphytic LAB, yeast, and mold were similar between sample 1 and sample 2. The count of LAB was lower than $10^3$ CFU/g FM, while yeast and mold counts were around $10^4$ CFU/g FM.

The characteristics of the selected LAB are shown in Table 2. *L. fermentum* 17SD-2 is Gram-positive, heterofermentation and catalase-negative and was able to grow at a wide range of pH (pH 3.5–9.0), a broad range temperature (15–45 °C), and a salinity of 6.5%. *L. fermentum* 17SD-2 used the following as carbon sources: ribose, galactose, D-glucose, D-fructose, D-mannose, maltose, lactose, melibiose, sucrose, D-raffinose, gluconate, and 5-keto-gluconate. Furthermore, this strain grew vigorously and possessed high FAE activity (Fig. 1).

## Effects of CE, LF, and LF+CE on the pH values and microbial characteristics of alfalfa silage

The final pH values and microbial populations of sample 1 and sample 2 were similar after ensiling (Table 3). Mold was not detected in all silage, and the significance analysis of main factors revealed that all additives (CE, LF, and CE+LF) significantly increased the LAB ($P < 0.001$) numbers, but decreased the yeast ($P < 0.001$) numbers. All treatments had lower ($P < 0.001$) pH values than those of control, particularly the lowest pH (4.6–4.7) in LF+CE treatments, which was much lower than the addition of CE or LF alone.

**Table 2  Characteristics of the selected strain Lactobacillus fermentum 17SD-2.**

| Characteristics | *Lactobacillus fermentum* 17SD-2 |
|---|---|
| Shape | Rod |
| Gram stain | + |
| Fermentation type | Heterofermentation |
| Catalase | − |
| Growth at pH | |
| 3.5 | + |
| 4.0 | + |
| 4.5 | + |
| 8.0 | + |
| 8.5 | + |
| 9.0 | + |
| Growth in NaCl | |
| 3.0% | + |
| 6.5% | + |
| Growth at temperature | |
| 15 °C | w |
| 20 °C | w |
| 25 °C | + |
| 30 °C | + |
| 35 °C | + |
| 40 °C | + |
| 45 °C | + |
| Fermentation of carbohydrate | |
| Ribose | + |
| Galactose | + |
| D-glucose | + |
| D-fructose | + |
| D-manose | + |
| Maltose | + |
| Lactose | + |
| Melibiose | + |
| Sucrose | + |
| D-raffinose | + |
| Gluconate | W |
| 5-keto-gluconate | + |

**Notes.**
+, positive; -, negative; W, weakly positive.

## Effects of CE, LF, and LF+CE on the fermentation quality and chemical composition of alfalfa silage

The fermentation parameters and chemical composition of the alfalfa silage after 60 days of ensiling are shown in Table 4. Lactic acid and acetic acid emerged as the main fermentation end products in all alfalfa silage. All additives remarkably increased the lactic acid content ($P < 0.001$), in particular, LF+CE had a significant effect on lactic acid and butyric acid

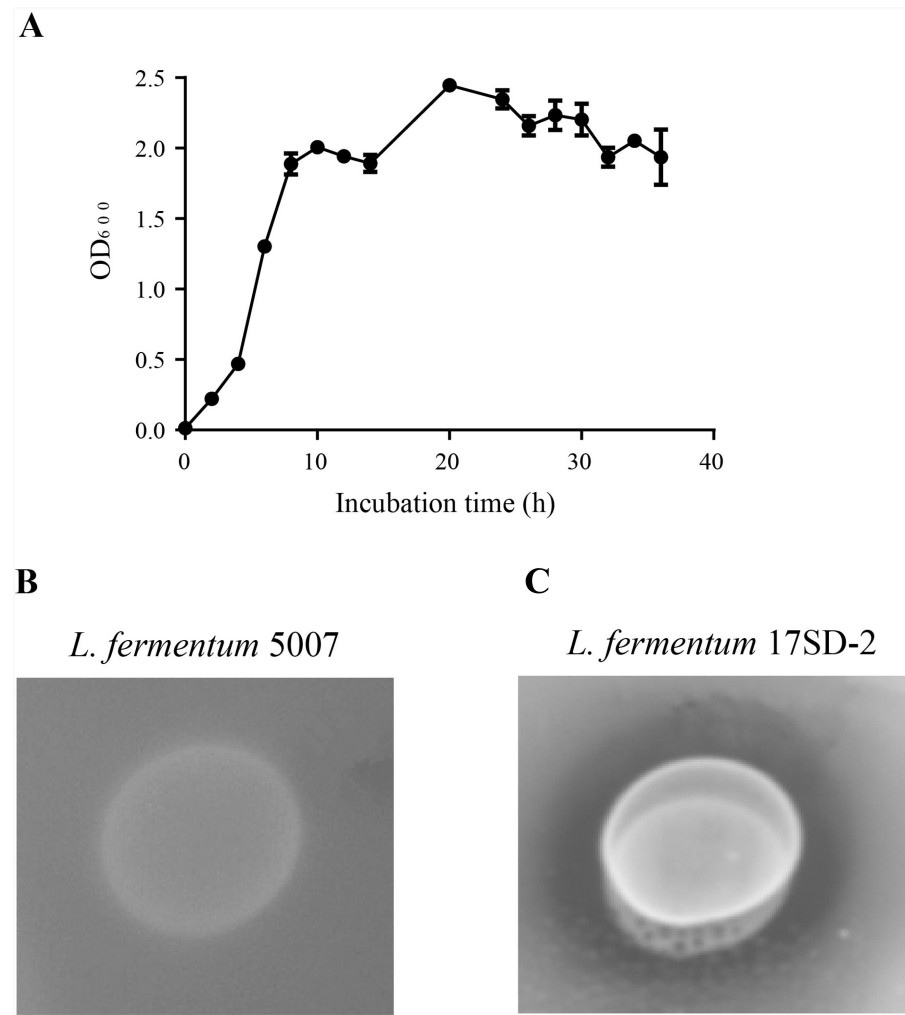

**Figure 1** **Growth curve and plate screening assay showing FAE activity by clear zone around the bacterial suspension.** (A) Growth curve of *Lactobacillus fermentum* 17SD-2. (B) *Lactobacillus fermentum* 5007 without FAE activity. (C) *Lactobacillus fermentum* 17SD-2 with FAE activity.

content. All treatments had higher lactic acid content and ratio of lactic acid and acetic acid, and LF+CE treatments had the highest lactic acid contents. In this study, all treated silage, especially LF+CE silage, had a lower level of butyric acid content (<1.0% DM), while a large amount of butyric acid (about 2.63% DM) appeared in the control silage of sample 1. In sample 2, no great difference was observed between control and additive treatments (except for LF+CE treatment).

Although all additives decreased ammonia-N ($P < 0.001$) and increased CP ($P < 0.001$) contents compared to control, the LF+CE treated silage had the highest contents of CP and lowest ammonia-N. Compared with the addition of CE alone, lower NDF and ADF contents were found in LF+CE treatment. Overall, the silage treated with FAE-producing LAB plus cellulase was better than those of the silage treated with either LAB or cellulase alone.

**Table 3  Cultivable microbial population and pH values of alfalfa silage after 60 days of ensiling.**

| Items | LAB log CFU/g FM | Yeast log CFU/g FM | Mold log CFU/g FM | pH value |
|---|---|---|---|---|
| **S1 group** | | | | |
| CK | $6.9 \pm 0.019^c$ | $4.6 \pm 0.0071$ | ND | $6.8 \pm 0.038^a$ |
| CE | $7.3 \pm 0.019^b$ | ND | ND | $5 \pm 0.026^d$ |
| LF | $7.3 \pm 0.015^b$ | ND | ND | $5.4 \pm 0.041^c$ |
| LF+CE | $7.4 \pm 0.019^b$ | ND | ND | $4.7 \pm 0.032^e$ |
| **S2 group** | | | | |
| CK | $6.8 \pm 0.024^c$ | $4.5 \pm 0.038$ | ND | $6.1 \pm 0.038^b$ |
| CE | $7.4 \pm 0.074^b$ | ND | ND | $5.1 \pm 0.018^d$ |
| LF | $7.6 \pm 0.046^a$ | ND | ND | $5.3 \pm 0.074^c$ |
| LF+CE | $7.4 \pm 0.014^b$ | ND | ND | $4.8 \pm 0.036^e$ |
| **SEM** | 0.049 | 0.019 | NA | 0.058 |
| **Main effects**[1] | | | | |
| WC | 0.132 | 0.052 | NA | <0.001 |
| A | <0.001 | <0.001 | NA | <0.001 |
| WC ×A | 0.003 | 0.019 | NA | <0.001 |

**Notes.**

LAB, lactic acid bacteria; FM, fresh matter; S1, Sample1 (fresh); S2, Sample 2 (wilted in the field for 5 h); CK, untreated control; CE, application of commercial cellulose; LF, application of L. fermentum 17SD-2; LF+CE, combination of L. fermentum 17SD-2 and commercial CE; SEM, Standard error of mean; ND, not detected; NA, Not applicable; 1. WC, water content; A, Additives; WC × A, the interaction between water content and additives.

## Effects of CE, LF, and LF+CE on microbial communities after 60 days of ensiling

High-throughput sequencing of 16S rRNA gene amplicons was conducted to systematically describe the bacterial communities in the raw material and silage. Alpha diversity of all samples is shown in Table 5. The coverage of all samples was above 0.99, which indicated that the sampling depth adequately captured most of the bacteria. Compared with sample 1, lower Chao and Shannon indexes were observed in sample 2, which was consistent with the results reported by *Wang et al. (2018)*, indicated that wilting (water content) had an important influence on silage microorganism. Lower observed species than pre-ensiled alfalfa crops and that of control appeared in CE alone or with LF treated silage in sample 2. In contrast, in sample 1, observed species in LF and LF+CE treated silages were higher, especially in LF treatment. Although we could not classify the behind reason clearly, probably related to the high number of the epiphytic detrimental microorganisms.

The dynamic variance of microbial population with different treatments can be demonstrated by principle co-ordinates analysis (PCoA). As shown in Fig. 2A (sample 1), component 1 and component 2 could explain 58.07% and 28.51% of the total variance, respectively; Similarity, in Fig. 2B (sample 1), component 1 and component 2 could explain 50.94% and 41%, respectively. Basically, Raw materials S1 and S2 could be well separated from all silage samples, indicating that ensiling was the main factor affecting anaerobic fermentation. Among silage samples, the distance between CK and CE treatments was near, similar phenomenon was observed between LF and LF+CE treatments, indicated

Su et al. (2019), *PeerJ*, DOI 10.7717/peerj.7712

Peerj

**Table 4 Fermentation characteristic and chemical composition in alfalfa silage.**

| Items | LA | AA | LA/AA | BA | DM | CP | NH$_3$-N | NDF | ADF | WSC |
|---|---|---|---|---|---|---|---|---|---|---|
| **S1 group** | | | | | | | | | | |
| CK | $2.8 \pm 0.13^d$ | $5.3 \pm 0.047^a$ | $0.52 \pm 0.028^d$ | $2.6 \pm 0.11^a$ | $20 \pm 0.094^e$ | $18 \pm 0.05^f$ | $1.8 \pm 0.014^a$ | $38 \pm 0.089^a$ | $30 \pm 0.095^a$ | $0.58 \pm 0.012^e$ |
| CE | $9.8 \pm 0.24^a$ | $5.3 \pm 0.057^a$ | $1.8 \pm 0.026^b$ | $1.2 \pm 0.011^b$ | $20 \pm 0.078^d$ | $21 \pm 0.15^e$ | $0.95 \pm 0.026^b$ | $32 \pm 0.023^d$ | $25 \pm 0.035^d$ | $0.64 \pm 0.0088^d$ |
| LF | $6.3 \pm 0.36^b$ | $4.7 \pm 0.048^b$ | $1.3 \pm 0.065^c$ | $0.84 \pm 0.06^d$ | $21 \pm 0.084^d$ | $21 \pm 0.066^d$ | $0.88 \pm 0.018^c$ | $34 \pm 0.068^b$ | $27 \pm 0.071^b$ | $0.52 \pm 0.015^f$ |
| LF+CE | $11 \pm 0.31^a$ | $4.2 \pm 0.039^c$ | $2.5 \pm 0.067^a$ | $1 \pm 0.068^c$ | $23 \pm 0.27^c$ | $25 \pm 0.05^a$ | $0.57 \pm 0.026^{de}$ | $28 \pm 0.038^g$ | $22 \pm 0.076^f$ | $0.98 \pm 0.015^b$ |
| **S2 group** | | | | | | | | | | |
| CK | $1.8 \pm 0.2^e$ | $2.9 \pm 0.023^f$ | $0.62 \pm 0.074^d$ | $0.57 \pm 0.018^e$ | $27 \pm 0.14^b$ | $22 \pm 0.043^c$ | $0.53 \pm 0.012^e$ | $33 \pm 0.084^c$ | $26 \pm 0.032^c$ | $1.2 \pm 0.018^a$ |
| CE | $5.9 \pm 0.38^b$ | $3.6 \pm 0.031^d$ | $1.6 \pm 0.099^b$ | $0.54 \pm 0.012^e$ | $27 \pm 0.24^b$ | $24 \pm 0.044^b$ | $0.59 \pm 0.019^d$ | $29 \pm 0.19^e$ | $23 \pm 0.09^e$ | $1.2 \pm 0.015^a$ |
| LF | $4 \pm 0.2^c$ | $3.2 \pm 0.058^e$ | $1.2 \pm 0.074^c$ | $0.58 \pm 0.028^e$ | $27 \pm 0.34^b$ | $22 \pm 0.077^c$ | $0.47 \pm 0.0063^f$ | $34 \pm 0.27^{bc}$ | $26 \pm 0.26^c$ | $0.79 \pm 0.0088^c$ |
| LF+CE | $6.7 \pm 0.21^b$ | $2.9 \pm 0.011^f$ | $2.4 \pm 0.083^a$ | $1 \pm 0.0028^c$ | $30 \pm 0.09^a$ | $25 \pm 0.061^a$ | $0.38 \pm 0.01^g$ | $29 \pm 0.16^f$ | $23 \pm 0.055^e$ | $1.2 \pm 0.0088^a$ |
| **SEM** | 0.38 | 0.06 | 0.097 | 0.073 | 0.27 | 0.11 | 0.025 | 0.2 | 0.16 | 0.018 |
| **Main effects**[1] | | | | | | | | | | |
| WC | <0.001 | <0.001 | 0.103 | <0.001 | <0.001 | <0.001 | <0.001 | <0.001 | <0.001 | <0.001 |
| A | <0.001 | <0.001 | <0.001 | <0.001 | <0.001 | <0.001 | <0.001 | <0.001 | <0.001 | <0.001 |
| WC ×A | <0.001 | <0.001 | 0.193 | <0.001 | 0.002 | <0.001 | <0.001 | <0.001 | <0.001 | <0.001 |

**Notes.**

LA, lactic acid; AA, acetic acid; LA/AA, the ratio of lactic acid and acetic acid; BA, butyric acid; NH 3 -N, ammonia-N; DM, dry matter; CP, crude protein; NDF, neutral detergent fiber; ADF, acid detergent fiber; WSC, water soluble carbohydrates; LAB, lactic acid bacteria; FM, fresh matter; S1, Sample1 (fresh); S2, Sample 2 (wilted in the field for 5 h); CK, untreated control; CE, application of commercial cellulose; LF, application of L. fermentum 17SD-2; LF+CE, combination of L. fermentum 17SD-2 and commercial CE; SEM, Standard error of mean; ND, not detected; NA, Not applicable; 1. WC, water content; A, Additives; WC × A, the interaction between water content and additives.

**Table 5  Alpha diversity of bacterial community after 60 days of ensiling.**

| Items | Observed species | Chao 1 | Shannon | Simpson | ACE | Coverage |
|---|---|---|---|---|---|---|
| **S1 group** | | | | | | |
| CK | 85 ± 5.6[b] | 110 ± 20[b,c] | 2.4 ± 0.084[a] | 0.15 ± 0.015[c,e] | 140 ± 34[a,b] | 0.99 |
| CE | 57 ± 3.5[c,d] | 100 ± 26[b,c] | 2.1 ± 0.014[b] | 0.17 ± 0.0044[c,d] | 120 ± 6.7[b] | 0.99 |
| LF | 130 ± 3.5[a] | 160 ± 6.8[a] | 2.4 ± 0.032[a] | 0.14 ± 0.0064[d,e] | 180 ± 9.7[a] | 0.99 |
| LF+CE | 85 ± 1.2[b] | 150 ± 25[a,b] | 2 ± 0.074[b] | 0.2 ± 0.016[c] | 180 ± 30[a] | 0.99 |
| **S2 group** | | | | | | |
| CK | 80 ± 3.8[b] | 100 ± 3.4[b,c] | 2.5 ± 0.084[a] | 0.12 ± 0.01[e] | 140 ± 7.4[a,b] | 0.99 |
| CE | 55 ± 3[c,d] | 75 ± 11[c] | 2 ± 0.037[b] | 0.18 ± 0.0063[c,d] | 83 ± 19[b] | 0.99 |
| LF | 66 ± 3.5[c] | 100 ± 6.5[b,c] | 1.2 ± 0.076[c] | 0.51 ± 0.031[b] | 130 ± 4.3[a,b] | 0.99 |
| LF+CE | 53 ± 3.8[d] | 70 ± 10[c] | 1.1 ± 0.036[c] | 0.58 ± 0.015[a] | 85 ± 2.5[b] | 0.99 |
| **SEM** | 5.2 | 22 | 0.085 | 0.021 | 26 | NA |
| **Main effects**[1] | | | | | | |
| WC | <0.001 | 0.001 | <0.001 | <0.001 | 0.003 | NA |
| A | <0.001 | 0.077 | <0.001 | <0.001 | 0.054 | NA |
| WC ×A | <0.001 | 0.237 | <0.001 | <0.001 | 0.093 | NA |

Notes.

S1, Sample1 (fresh); S2, Sample 2 (wilted in the field for 5 h); CK, untreated control; CE, application of commercial cellulose; LF, application of L. fermentum 17SD-2; LF+CE, combination of L. fermentum 17SD-2 and commercial CE; SEM, Standard error of mean; ND, not detected; NA, Not applicable; 1. WC, water content; A, Additives; WC × A, the interaction between water content and additives.

that the addition of *L. fermentum* 17SD-2 had a significant effect on bacterial community in alfalfa silage. In addition, the variation in microbial community might be one critical factor leading to difference in silage quality.

The bacterial communities at the genus level of all alfalfa silage before and after 60 days of ensiling are shown in Figs. 2C and 2D. Before ensiling, in both raw materials (S1 and S2), *Cyanobacteria* and *Pantoea* were the most abundant genera, above 53.86% and 8.33% of the entire community, respectively. However, after fermentation, the high abundance of *Cyanobacteria* and *Pantoea* significantly decreased or even disappeared. *Enterobacter and Lactobacillus* gradually became the most prevalent genera. Although *Enterobacter* was detected in all silage, their relative abundance decreased from 49.98% to 17.01% with the addition of LF and CE, furthermore, LF alone or with CE treated silage showed much lower *Enterobacter* abundance than control and CE alone treatment. In addition, a similar phenomenon was observed in sample 2.

*Lactobacillus*, a desirable genus, was present in all silage and *Lactobacillus* was highly abundant in LF or LF+CE-treated silage, especially in sample 2, where this genus accounted for 85.36% and 85.50% of the total community, respectively. Many other LAB genera, i.e., *Pediococcus*, *Enterococcus*, *Weissella*, and *Lactococcus,* were also detected after ensiling. For example, in sample 2 silage, the proportions of *Pediococcus* and *Enterococcus* ranged from 10.15% to 1.26% and 11.35% to <0.1%, respectively. *Weissella* and *Lactococcus* were mainly detected in control and CE treatment. Additionally, relatively high proportions of *Anaerosporobacter* (5.96%), *Clostridium* (4.78%), and *Garciella* (3.65%) were observed in the control of sample 1.

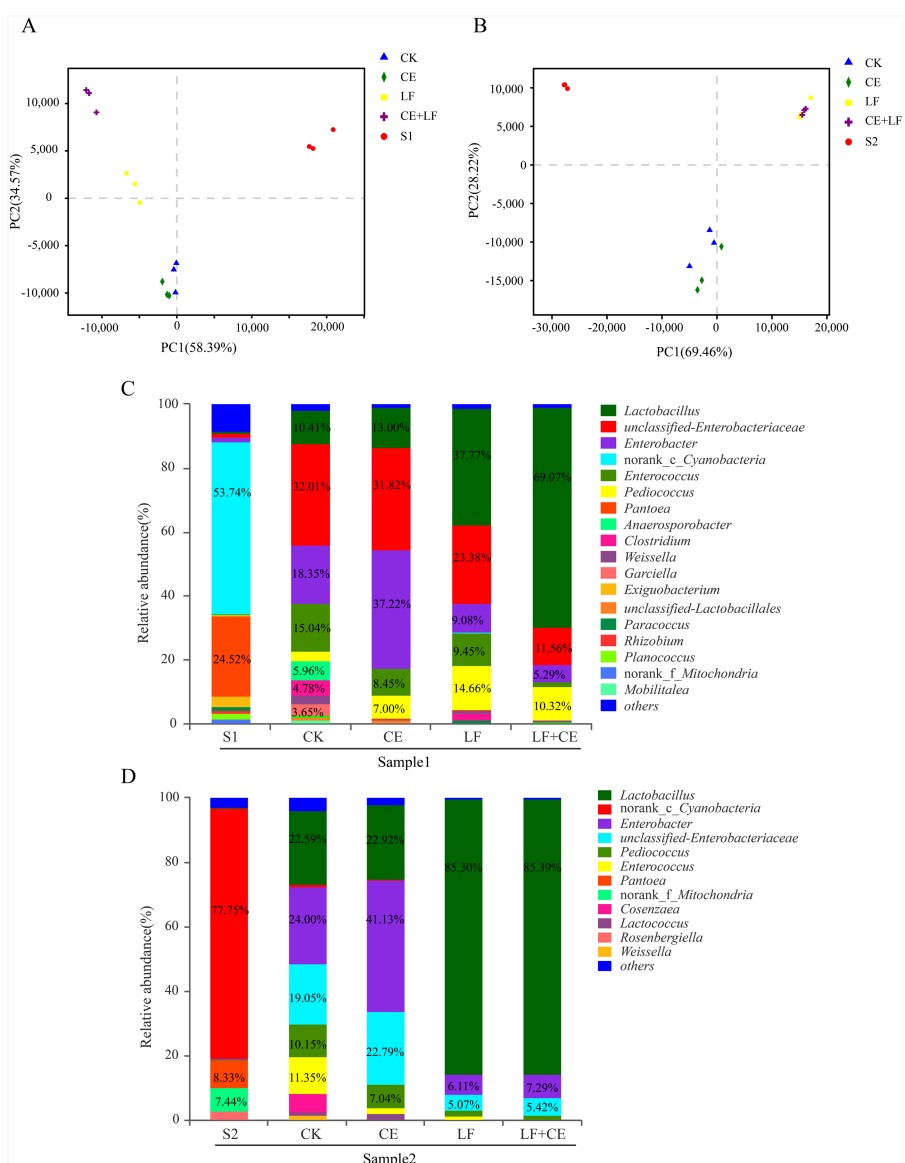

**Figure 2** **PCoA analysis and relative abundance of bacterial community at the genus level before and after ensiling.** (A) PCoA analysis in sample 1; (B) PCoA analysis in sample 2; (C) Relative abundances of bacterial composition at genus level in sample 1; (D) Relative abundances of bacterial composition at genus level in sample 2. S1, pre-ensiled sample 1; S2, pre-ensiled sample 2; CK, control (without any additives); CE, cellulase; LF, *Lactobacillus fermentum* 17SD-2; LF+CE, combination of *Lactobacillus fermentum* 17SD-2 and cellulase.

## Relationship between main genera and silage quality

To better understand the relationships between bacterial compositions and silage properties, a Spearman correlation heatmap was constructed at the genus level (Fig. 3). Spearman correlation analysis illustrated that *Lactobacillus* was positively correlated with LA/AA ($r = 0.47$), DM ($r = 0.69$), and CP ($r = 0.71$) contents, while it was negatively correlated with concentration of ammonia-N ($r = -0.81$) and pH ($r = -0.47$).

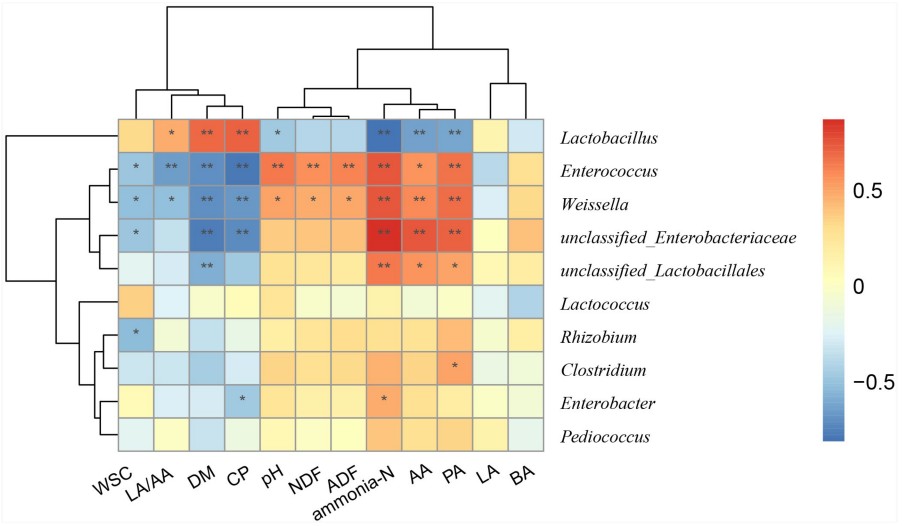

**Figure 3** **Spearman correlation heatmap between the main genera and silage quality.** R was presented in different colors; the right side of the legend is the color range of different R values. The value of $P \leq$ 0.05 is marked with "*", $P \leq 0.01$ is marked with "**".

*Enterococcus* and *Weissella* were negatively correlated with LA/AA ($r = -0.65$ and $-0.51$) and positively with pH ($r = 0.64$ and 0.51) and ammonia-N ($r = 0.75$ and 0.75). *Enterobacter* was positively correlated with the concentration of ammonia-N ($r = 0.47$), but was negatively correlated with CP ($r = -0.47$) content.

## DISCUSSION

Although ensiling is considered an effective method for preserving high moisture forage, it has shown some limitations in the production of high-quality alfalfa silage due to its low WSC content and high buffering capacity. During the ensiling process, the epiphytic LAB naturally present on forage crops are responsible for silage fermentation and also influence silage quality, which mainly transforms WSC into organic acids, primarily lactic acid, to inhibit the activity of undesirable bacteria. However, in the present study, low WSC contents (below 1.80% DM), a low number of epiphytic LAB ($<10^3$ CFU/g of FM) and high buffering capacity may lead to unsatisfactory preservation. Therefore, the ensiling process should be regulated by additions such as cellulase and LAB. In the present study, the selected *Lactobacillus fermentum* 17SD-2 strain grew well at low pH and it could grow at road temperature. Additionally, this strain grows vigorously and has high FAE activity.

FAE, an important auxiliary enzyme, plays an essential role in plant cell wall hydrolysis by breaking the ester-linked bond between cell wall phenolic acids and polysaccharides, which makes the plant cell wall more accessible to enzyme attack (*Faulds, Perez-Boada & Martinez, 2011*). Recently, researchers have increasingly reported that most commercial enzyme additives are already mixtures of several fibrolytic enzymes and the effect of these additives on silage chemical composition is actually an integrated effect of synergistic interactions (*Li et al., 2018*; *Li et al., 2014*).

As expected, in the current study, compared with addition of CE alone, lower NDF and ADF contents were found in FAE-producing *L. fermentum* 17SD-2 and cellulase (LF+CE) treatment, and a similar result was found in mixed small-grain silage (*Jin et al., 2017*). These results indicated that the combination of LF and CE might have a synergistic effect, while whether the fiber degradation rate *in vivo* is higher in LF+CE group than the other two (LF and CE) groups need further research. However, it has been reported that the use of an enzyme product in combination with a FAE-producing inoculant did not improve corn silage fermentation or its nutritive value (*Lynch, Baah & Beauchemin, 2015*). This discrepancy might be caused by the abundantly epiphytic LAB population of corn compared to alfalfa, resulting in inoculated FAE-producing LAB not dominating the epiphytic microbial population at the ensiling process of corn.

The pH value is considered an important indicator reflecting microbial activity and silage fermentation. In this study, all treatments had lower pH values but higher lactic acid content than those of control. The relatively lower pH value and higher content of lactic acid in the CE and LF+CE-treated silage, especially the pH values in LF+CE, declined to around 4.6–4.7, much lower ($P<0.05$) than that of the addition of CE or LF alone, which could be due to the direct increase of the fermentable substrate through degrading the plant cell wall. All treated silage had a higher ratio of lactic acid and acetic acid, which might indicate that the silage fermentation of alfalfa can be driven towards a homo-fermentative model with the addition of CE and LF.

Additionally, for well-preserved silage, butyric acid content is less than 1.0% DM (*McDonald, Henderson & Herson, 1991*) and is usually produced by *Clostridia*. For instance, in sample 1 silage, all treatments, especially LF+CE treatment, had a lower level of butyric acid content (<1.0% DM). Control silage had unsatisfactory preservation due to low content of lactic acid, high pH, and high content of butyric acid and ammonia-N together with high abundance of *Clostridium*, *Anaerosporobacter*, and *Garciella* (Fig. 2C). These bacteria are obligate anaerobes from the class *Clostridia*. *Clostridia* is an influential component of bacterial communities and is often detected in grass silages. The appearance of *Clostridia* is undesirable because they can take advantage of protein and water soluble carbohydrate to produce butyric acid and consequently affect silage quality due to unpleasant odor (*Zheng et al., 2017*). That may help explain the poor fermentation quality in the control of sample 1, which was consistent with the report by *Nishino et al. (2012)*. However, a similar result was not observed in sample 2 silage, which was possibly because high DM content could efficiently inhibit the growth of *Clostridium*.

The ammonia-N level reflects the CP degradation in silage, which is an important criterion for evaluating silage quality. Although all additives decreased ammonia-N but increased CP content compared with those of control, especially, the LF+CE treated silage had the highest content of CP and lowest ammonia-N, and similar results were also found in *Stylo* silage produced using a mixture of LAB and cellulase (*Li et al., 2017*). The most plausible reason was that the LF+CE addition could efficiently inhibit the growth of undesirable proteolytic bacteria, such as *Enterobacter*. *Enterobacter* is generally considered undesirable during the ensiling process because it can ferment lactic acid to acetic acid and other products, thus subsequently causing nutrition loss (*Ni et al., 2017*). In this

study, although *Enterobacter* was detected in all silage, their relative abundance was much lower in the LF and LF+CE treated silage. In contrast, desirable LAB such as *Lactobacillus*, *Enterococcus*, *Pediococcus*, and *Weissella*, rod-shaped *Lactobacillus* plays a critical role in enhancing lactic acid content and reducing pH value (*Cai et al., 1998*). In the present study, *Lactobacillus* was the most abundant genus in LF or LF+CE treated silage, which might account for their relatively high fermentation quality.

## CONCLUSION

Compared with addition of CE or LF alone, inoculation of a FAE-producing lactic acid bacteria in combination with cellulase not only improved the silage fermentation quality of alfalfa, but also inhibited the growth of undesirable microbes. Silage inoculants consisting of FAE-producing LAB might be an effective way to improve the silage quality of alfalfa.

### Funding

This work was supported by grants from the Special Fund for Agro-scientific Research in the Public Interest (201503134) and the National Natural Science Foundation of China (31570114). The funders had no role in study design, data collection and analysis, decision to publish, or preparation of the manuscript.

### Grant Disclosures

The following grant information was disclosed by the authors:
Special Fund for Agro-scientific Research in the Public Interest: 201503134.
The National Natural Science Foundation of China: 31570114.

### Competing Interests

Chen Jie and Liu Yan are employed by Hebei Zhongyuzhongke Biotechnology Development Company.

### Author Contributions

- Rina Su conceived and designed the experiments, performed the experiments, analyzed the data, contributed reagents/materials/analysis tools, prepared figures and/or tables, authored or reviewed drafts of the paper, approved the final draft.
- Kuikui Ni conceived and designed the experiments, analyzed the data, contributed reagents/materials/analysis tools, prepared figures and/or tables, authored or reviewed drafts of the paper, approved the final draft.
- Tianwei Wang analyzed the data, contributed reagents/materials/analysis tools, prepared figures and/or tables, authored or reviewed drafts of the paper, approved the final draft.
- Xiaopan Yang and Jie Zhang analyzed the data, contributed reagents/materials/analysis tools, prepared figures and/or tables, approved the final draft.
- Yayong Liu approved the final draft.
- Weixiong Shi performed the experiments, approved the final draft.
- Liu Yan and Chen Jie performed the experiments.
- Jin Zhong conceived and designed the experiments, analyzed the data, authored or reviewed drafts of the paper, approved the final draft.

## Data Availability

Data is available at NCBI SRA: SRP189118, PRJNA528535.

## Supplemental Information

Supplemental information for this article can be found online at http://dx.doi.org/10.7717/peerj.7712#supplemental-information.

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
