# Peer review of "Effects of ferulic acid esterase-producing Lactobacillus fermentum and cellulase additives on the fermentation quality and microbial community of alfalfa silage"

_PeerJ, doi:10.7717/peerj.7712_

## Round 0.1 · original submission · Major Revisions

Both reviewers have comments on the statistical analysis on your data. Those need to be addressed carefully in the revised version of the manuscript.

Reviewer 1 ·

Basic reporting

In most parts, the manuscript is clear and is well-structured

Experimental design

Reflection on the aim of study:
FAE producing inoculants are used to open up complex structure of plant fiber during ensiling to improve ruminal degradation of fiber. It is not common to apply them to provide substrates (with helps of fibrolytic enzymes) for a successful ensiling. This approach might actually have adverse effect on fiber digestibility in the rumen, something which could be speculated from the results obtained (see below)

Sampling:
Maturity stage of alfalfa should be mentioned. In addition, it is informative to provide data about size of the field and distance between the two sampling points.

Methods:
For DM determination, samples should be dried at 103°C. Drying at 65°C will not remove all the sample water. In addition, for silage samples, DM should be corrected for losses of volatiles during drying.

Statistical analysis:
The experiment comprises two factors: sample (sample 1 and sample 2) and treatment and therefore one-way ANOVA should not be used. Alternatively, I suggest to use mixed model and include sample as random factor. In addition, the method used for pairwise comparison should be mentioned.

Validity of the findings

Reflection on the results:
To evaluate the effect of treatments on DM loss during ensiling, I suggest estimating losses of DM during ensiling (i.e. DM in –DM out) instead of only presenting silage DM.

The NDF represents fiber fraction of the biomass. The reduction of NDF during ensiling by LF+CE treatment suggests that this treatment would likely reduce ruminal degradation of fiber. This is because easily degradable parts of the fiber, which are otherwise utilized by rumen microbes, are already degraded and utilized during ensiling, increasing the concentration of low degradable parts of the fiber. In order to confirm /reject this theory, silage samples can be subjected to in vitro ruminal degradation assays.

Reviewer 2 ·

Basic reporting

This article is well written in English. However, sufficent introduction and background should be include to demonstrate. For example, manyprevious experiments have been conducted to investigate the effects of FAE-producing LAB on silage quality and animal performance, and you should introduce recent findings, main opinions and shorting points of FAE-producing inoculants, and explain why you selected L. fermentum as your silage additives.
Authors should provide the raw data of bacterial diversity, espcially for high-throughput sequencing.Data on alpha diversity of bacterial community after 60 days of ensiling was not well/statistically shown in Table 5. The same question on relative abundance of certain genus such Lactobacillus, occurred in Figure 2.
In addition, a hypothesis is necessary for your introduction.

Experimental design

more details were for morphological, physiological, and biochemical characteristics of L. fermentum17SD-2. In addition more details for experimental design and samplings will help our understanding. Please note the true replicate in experimental design, especially for microbial diversity analysis.

Validity of the findings

no comment

Additional comments

This research is helpful to develop effective LAB inoculants to enhance fermentation of high fiber-contained silage. However, authors will need to revise the manuscript, if published in Peer J. The more concerning issues are as following:

ABSTRACT
More silage fermentation parameters such as pH, ammonia-N and volatile fatty acids were shown in abstract section.

INTRODUCTION
This article should include more relevant literature for sufficient introduction and background. Author should introduce the recent finds, main opinions and shorting points of FAE-producing inoculants, and explain why you selected L. fermentum as your silage additives. In addition, a hypothesis is necessary for your introduction.
L67-68, an opposite conclusion from their study was that inoculation of FAE-producing LAB did not alter the growth performance of finishing feedlot cattle.
L70, wrong reference,the inoculants used in the study of Chen et al (2017) was not FAE-produceing LAB.
L73, more references were necessary, such as Li et al (2019), Lynch et al (2014), Jin et al (2015), Addah et al (2014), etc.
L77, the objectives of this study is to (1) characterize FAE-producing L. fermentum (LF) and (2) investigate the effects of LF, celluase or their combination on silage fermentation and bacterial community?

MATERIALS AND METHODS
In this section, more details were for morphological, physiological, and biochemical characteristics of L. fermentum17SD-2. In addition more details for experimental design and samplings will help our understanding.
L88-89, explained why choose L. plantarum TS15.2 not other no FAE-producing L. fermentum as negative control.
L90, the diameters of a clear zone was ?
L91, delete the reference of Xu et al. (2017)
L100, why not add growth temperature of <20℃. If not, the description of wide growth range is not appropriate.
L106-107, how did you gain the two DM contents, by wilting or advanced maturity stage? And, each sample as group was randomly?
L117-119, more details for experiment design. Provide true replicates.
L149, samples for microbial diversity analysis was from the three replicates with some treatment?
L161, is it bacterial diversity analysis?
L175, data on alpha diversity of bacterial community after 60 days of ensiling was not well/statistically shown in Table 5. The same question on relative abundance of certain genus such Lactobacillus, occurred in Figure 2.

RESULTS AND DISCUSSION
It is well known that alfalfa is not easily ensiled for higher buffer capacity and relatively lower WSC content as compared with grass. In fact, LAB dominated microbial population in control silage. Meanwhile, the count of yeasts was below 5.0 log10 cfu/g FM. Why author concluded that poor fermentation resulted from low LAB counts, but not from high buffer capacity? Author should focus on the CP contents between treatments. Importantly, author should explain why LF+GE increased richness in silages from sample 1, but decreased that from sample 2. Moreover, why the bacterial diversity of ensiled alfalfa was higher than that of fresh alfalfa? Importantly, explain how chemical compositions such as WSC, DM, NDF and NDF affected Lactobacillus, Enterococcus and Weissella.

Others
Aata on alpha diversity of bacterial community after 60 days of ensiling was not well/statistically shown in Table 5. The same question on relative abundance of certain microorganism such Lactobacillus, occurred in Figure 2. The relative abundance of top 10 genus should be included, and that of certain microorganism such Lactobacillus should be statistically illustrated in Figure 2 C and D. If possible, please change the Figure A and B as PCoA.

Annotated reviews are not available for download in order to protect the identity of reviewers who chose to remain anonymous.

---

## Round 0.2 · Minor Revisions

There are still concerns of one reviewer you should try to address adequately. In particular, you should discuss the digestibility of your silage products (you could consider to refer to other groups' work or future studies).

Reviewer 1 ·

Basic reporting

Line 99-105: Explain how the evaluations were performed: (e.g. how did you measure salt tolerance or how did you measure bacterial growth?)

Line 123: The sentence “Each sample as group was randomly” is linguistically wrong.

Line 148: Did you correct silage DM for losses of volatile during drying?

Line 148: The sentence “DM contents and ground to pass a 1.0 mm screen for chemical analysis” is linguistically wrong.

Line 155-159: Why did you centrifuge the bacterial suspension twice?

Line 199: Replace “indigenous” with “epiphytic”

Line 305 and 350: Degradation of fibre during ensiling is not necessarily a good thing. See pubmed: 11063325

Figure 1: Stated in the method and material that bacterial growth was measured by measuring OD.

Table 3, 4 and 5: All the abbreviations should be defined. In addition, the values of main factors together with SEM should be shown.

Experimental design

No comments

Validity of the findings

The study provides useful information about the effect of feruloyl esterase producing inoculant and cellulase and their combination on fermentation quality of alfalfa silage. Unfortunately, the author did not provide information as to how the treatments affected the digestibility of silage. It is important to consider that improving fermentation quality of silage at the expense of silage digestibility is not desirable.

The author replied in the rebuttal that in vitro evaluation of silage digestibility was not in the scope of the study. However, without this information the paper is incomplete in my view. Therefore, as it is I cannot recommend this article for publication.

Reviewer 2 ·

Basic reporting

No comment

Experimental design

no comment

Validity of the findings

no comment

Additional comments

The manusctipt "Effects of ferulic acid esterase-producing Lactobacillus fermentum and cellulase additives on the fermentation quality and microbial community of alfalfa silage" (#35821) has been revised well.

---

## Round 0.3 · accepted · Accept

The authors addressed the additional comments of the reviewers. Although no experimental data on digestibility is presented, the obtained results are discussed adequately.

#